# Analysis of the Clinicopathological Characteristics, Prognosis, and Lymphocyte Infiltration of Esophageal Neuroendocrine Neoplasms: A Surgery-Based Cohort and Propensity-Score Matching Study

**DOI:** 10.3390/cancers15061732

**Published:** 2023-03-13

**Authors:** Long Zhang, Boyao Yu, Zhichao Liu, Jinzhi Wei, Jie Pan, Chao Jiang, Zhigang Li

**Affiliations:** 1Department of Thoracic Surgery, Section of Esophageal Surgery, Shanghai Chest Hospital, Shanghai Jiao Tong University School of Medicine, Shanghai 200030, China; 2Department of Pathology, Shanghai Chest Hospital, Shanghai Jiao Tong University School of Medicine, Shanghai 200030, China

**Keywords:** neuroendocrine neoplasms, esophageal carcinoma, tumor-infiltrating lymphocytes, clinicopathological features, prognosis, immunotherapy

## Abstract

**Simple Summary:**

Esophageal neuroendocrine neoplasms (E-NENs) are a rare malignancy in esophageal carcinoma; their clinical and oncologic characteristics are poorly reported. Additionally, the effect of surgery on E-NENs remains unclear and controversial. In this study, we retrospectively analyzed the clinicopathological characteristics, prognosis, and immune cell infiltration in E-NENs and compared them with those of esophageal squamous cell carcinoma (ESCC) to determine whether surgery has the same therapeutic efficacy for E-NENs as for ESCC based on a cohort who received surgical treatment. According to our results, first, the target population for surgery may be limited to stage I E-NENs patients. Secondly, E-NENs and especially pure NENs were correlated with the cold tumor phenotype because of the less infiltration of immune cells compared with ESCC. The findings suggest that strategies for immune activation should be developed and applied when immunotherapy is considered for E-NENs in the future.

**Abstract:**

Background: Esophageal neuroendocrine neoplasms (E-NENs) are a rare and poorly reported subtype of esophageal carcinoma. We analyzed the differences in clinicopathological features, prognosis, and tumor-infiltrating lymphocytes (TILs) between E-NENs and esophageal squamous cell carcinoma (ESCC). Methods: A total of 3620 patients who underwent esophagectomy were enrolled retrospectively. The study cohort was divided into two groups (E-NENs and ESCC) through propensity-score matching, and the prognosis and TILs were compared between the two groups. The TILs were assessed using tumor specimens (including six cases of ESCC, six cases of neuroendocrine carcinomas [NECs], and six cases of mixed neuroendocrine–non-neuroendocrine neoplasms [MiNENs]). Results: E-NENs accounted for 3.0% (107/3620) of cases, among which there were just 3 neuroendocrine tumor cases, 51 NEC cases, and 53 MiNENs cases. After matching, esophageal neuroendocrine carcinomas (E-NECs) showed both poorer 5-year overall survival (OS; 35.4% vs. 54.8%, *p* = 0.0019) and recurrence-free survival (RFS; 29.3% vs. 48.9%, *p* < 0.001) compared with ESCC. However, the differences were not prominent in the subgroup with stage I. No significant survival benefit was observed for E-NECs with multimodal therapy. Multivariate analysis demonstrated that E-NECs are an independent risk factor for OS and RFS. In the exploratory analysis, E-NECs were associated with less infiltration of immune cells compared with ESCC. Conclusion: E-NECs are significantly associated with a poorer prognosis than ESCC except for early-stage disease. The fewer TILs within the tumor microenvironment of E-NECs compared with ESCC results in weaker anti-tumor immunity and may lead to a poorer prognosis.

## 1. Introduction

Esophageal carcinoma (EC) is the seventh most common cancer and the sixth leading cause of cancer death worldwide [1]. Squamous cell carcinoma (SCC) is the most common tissue type of EC, followed by adenocarcinoma (AC) and other subtypes. Neuroendocrine neoplasms (NENs) originate from the neuroendocrine system and are commonly found in digestive tract and lung tumors [2]. Esophageal NENs (E-NENs) are a rare subtype of EC with poorly reported clinical and oncologic characteristics. E-NENs are characterized by significant heterogeneity, malignancy, and a poor prognosis [3]. The diagnosis of NENs relies on cell morphology and specific biomarkers such as chromogranin A (CgA) and synaptophysin (Syn) [4]. With the improvement of early tumor screening and diagnostic techniques, the incidence of NENs is gradually increasing. The annual incidence of NENs has risen from 1.09 per 10,000 in 1973 to 6.98 per 10,000 in 2012 [5].

The standard treatment for esophageal squamous cell carcinoma (ESCC) is multimodal therapy based on surgery, with a five-year overall survival (OS) close to 50–60% [6,7]. Meanwhile, patients with distant disease are mainly treated by systematic therapies such as chemotherapy (CT). However, there are no authoritative guidelines or consensus on the treatment of E-NENs owing to the lack of cases and clinical studies. The treatment methods for E-NENs mainly include surgery, CT, radiotherapy (RT), interventional therapy, and biological therapy [8]. Among them, radical surgery is an important treatment method. However, the effect of surgery for esophageal neuroendocrine carcinomas (NECs) remains controversial. Situ et al. reported that radical esophagectomy should be the first option for limited-stage small cell NECs [9]. In contrast, Sohda et al. [10] reported a non-obvious effect of surgery for E-NECs [10]. For gastrointestinal NENs, curative resection is the final goal that should be pursued as far as possible. Therefore, in the context of an increasing number of cases, it is important to explore whether surgery has a survival benefit in E-NENs patients.

Unfortunately, due to histological rarity, there is little research on E-NENs. Moreover, almost no reports have described the differences in oncology features, prognosis, and immune cell infiltration between ESCC and E-NENs. Therefore, in the present study, we compared the clinicopathological characteristics, prognosis, and lymphocyte infiltration between ESCC and E-NENs and investigated whether surgery has the same therapeutic efficacy in the two subtypes. The findings are in the hope of providing valuable clinical guidance for the treatment of E-NENs.

## 2. Methods

### 2.1. Study Cohort

This was a single-center (Department of Thoracic Surgery, Shanghai Chest Hospital, Shanghai, China) retrospective study of all patients who underwent esophagectomy and were histologically diagnosed with ESCC or E-NENs from 2012 to 2021. Our center is a high-volume referral unit involved in the multimodal management of EC [11]. Patients who did not obtain R0 resection or had incomplete follow-up and clinicopathological data were excluded. Patients with distant disease or who died within the first 30 days or in the hospital were also excluded. Patients, where the primary tumor was identified as cancer in situ or high-grade intraepithelial neoplasia, were excluded. Patients who received radical chemoradiotherapy were not included in this study (see the flowchart in Figure 1). This study was approved by the ethics committee of Shanghai Chest Hospital.

The main endpoints of this study were OS, recurrence-free survival (RFS), and clinicopathological characteristics for E-NENs and ESCC patients. We also quantified the main immune cells of ESCC and E-NECs in an exploratory analysis.

### 2.2. Preoperative Examination

All enrolled patients received normative preoperative assessments, including physical examination, standard laboratory tests, and imaging examination. Endoscopy with biopsy was used for diagnosis. For tumors suspected of invading the trachea, an ultrasound bronchoscope was used to determine the invasion depth. Contrast-enhanced computed tomography of the neck, chest, and abdomen is the most commonly used examination to evaluate tumor stage. Others include ultrasound of neck and positron emission tomography-computed tomography (PET-CT), which is applied only if clinically needed.

### 2.3. Surgery and Perioperative Treatments

According to the location and stage of the tumor, different surgeries are recommended for patients, including right (McKeown, Ivor–Lewis) or left (Sweets) transthoracic approach surgery. The surgical procedures consisted of transthoracic esophagectomy, stomach mobilization, and standard 2-field lymphadenectomy. Surgical procedures included conventional open surgery and minimally invasive surgery. Neoadjuvant or adjuvant therapy, typically including CT, RT, and immunotherapy, are recommended for patients with locally advanced tumor stages [12]. The chemotherapy regimens of platinum combined with paclitaxel were used in neoadjuvant and adjuvant therapy. The dose of radiotherapy was recommended for the patients according to guidelines [12]. Moreover, immunotherapy combined with chemotherapy as neoadjuvant treatment was applied to partial patients.

### 2.4. Diagnosis and Classification

The diagnosis of histology mainly depends on the gross specimens, while tumor markers are used to assist in the diagnosis. In the present study, tumor specimens were fixed in formalin, embedded in paraffin, sectioned, and stained with hematoxylin and eosin (H-E). All sections were reviewed and analyzed by senior pathologists. If the tumor cell morphology was in accordance with E-NENs, specific markers, including CgA, Syn, CD56, and Ki-67, were assessed (see Appendix A).

NENs were classified and graded in accordance with the World Health Organization (WHO) classification of Tumors of the Digestive System (2019 version) [13]. According to the tumor cell differentiation and proliferation, E-NENs were classified into the following categories: NETs, neuroendocrine carcinomas (NECs), and mixed neuroendocrine–non-neuroendocrine neoplasms (MiNENs). NETs, when well differentiated, were graded into G1, G2, and G3 based on mitotic number and Ki-67 proliferation index according to the WHO classification 2019 [13]. All NECs belong to poorly differentiated, high-grade tumors, whereas the differentiation and grading of MiNENs is undetermined. Finally, the stage of E-NENs was determined according to the American Joint Committee on Cancer staging system for EC (eighth edition).

### 2.5. Outcome Follow-Up

All enrolled patients were examined in the outpatient clinic. Examination was performed every three months for the first two years, with reviews every six months after that. The median follow-up was 36.5 months (interquartile range, 24.0–60.9), and the latest follow-up was 31 May 2022. Considering the difference in biological behavior and small sample size, the 3 cases of NETs were not included in survival analysis. For the patients who received upfront surgery, OS was defined as the time from the date of surgery to death. RFS was defined as the time from the date of surgery to relapse. However, OS and RFS of the patients who received neoadjuvant therapy was defined as the time from the date of initial diagnosis to death and relapse, respectively. In cases of suspected recurrence, computed tomography, PET-CT, upper gastrointestinal endoscopy with biopsy, and ultrasonography were performed as needed.

### 2.6. Assessment of Tumor-Infiltrating Lymphocytes (TILs)

The TILs of E-NENs and ESCC were identified using immunohistochemical staining (IHC). All gross specimens were from patients without preoperative treatments. First, the tumor specimens were fixed in formalin, embedded in paraffin, and sectioned. Second, 4 μm-thick tissue sections were dewaxed and hydrated using xylene and graded ethanol. The sections were then incubated in a water bath with sodium citrate buffer at 98 °C for 20 min and treated with 3% hydrogen peroxide and methanol for 30 min to depress endogenous peroxidase. The sections were incubated at 4 °C overnight after the addition of primary antibodies. Secondary antibodies were added the next day, and the sections were incubated at 37 °C for 1 h. Finally, after adding diaminobenzidine chromogen, the sections were stained with hematoxylin, dehydrated with ethanol, cleared with xylene, and sealed with neutral resin. (All product information is presented in Appendix A).

We randomly selected paraffin sections of six cases with pure neuroendocrine carcinomas (pNECs), NENs mixed with squamous cell carcinoma (NmSCC), and ESCC. First, using H-E staining, we distinguished the tumor nest (TN) and the tumor stroma (TS) regions and then assessed the number of TILs, including CD4^+^ and CD8^+^ T cells and CD20^+^ B cells. After the IHC detection of TILs, three random fields of view under 20× magnification were selected using an optical microscope for each case, and the mean number of positive lymphocytes in the three fields was summed and divided by the field size. The CD4^+^ and CD8^+^ T cells in the TN and TS were counted separately from the B cells. The mean numbers of the three types of cells were then compared.

### 2.7. Statistical Analysis

Baseline characteristics included categorical and continuous variables. To compare differences between the two groups, the χ^2^, *t*, and non-parametric test were used for categorical and continuous variables. The same method was used for the comparison of the number of TILs. Discrepancies in the baseline characteristics between the two groups were balanced using propensity-score matching. The matching was performed 1:1 with a caliper value of 0.07, and the matching effect was favorable (see Appendix A). OS and RFS after matching were calculated and plotted using the Kaplan–Meier method, and survival curves were compared using log-rank test. The hazard ratios of the variables for OS and RFS were calculated using the Cox proportional hazards model, including before-and-after matching. A two-sided *p* value of <0.05 was deemed statistically significant. Statistical analysis was performed using SPSS 25.0 software (IBM Corporation, Armonk, NY, USA).

## 3. Results

### 3.1. Overall Cohort Characteristics

Between January 2012 and December 2021, a total of 3620 patients were enrolled in the study. There were 3513 (97.0%) ESCC cases and 107 (3.0%) E-NEN cases. The median age was 64 years. The majority of cases in both groups were male. A total of 69.8% (2527/3620) of patients received minimally invasive esophagectomy (92.5% [99/107] E-NENs and 69.1% [2428/3513] ESCC). The proportion of patients who received neoadjuvant or adjuvant therapies was 56.3% (2037/3620), including 68.2% (73/107) for E-NENs and 55.9% (1964/3513) for ESCC. When the discrepancies between the two groups were balanced through propensity-score matching, a matched cohort emerged. The demographic and baseline characteristics of the study population are shown in Table 1.

### 3.2. Pathological Outcomes

Similar to ESCC, the tumor location of E-NENs was dominated by the middle (*n* = 51, 47.7%) and lower (*n* = 49, 45.8%) esophagus. Pathological analysis revealed a significant difference between the two groups in the tumor-invasive depth (E-NENs, T3 + T4 = 36.4% vs. ESCC, T3 + T4 = 55.1%; *p* < 0.001). Lymph-node metastasis (63.5% vs. 46.9%, *p* < 0.001) and advanced tumors (stages III + Iva, 52.3% vs. 43.9%, *p* = 0.011) were more common in E-NENs than in ESCC. In addition, the vast majority of E-NENs were poorly differentiated tumors except for three cases of NETs, which were well differentiated. Lymphovascular invasion was more common in E-NENs than in ESCC (27.1% vs. 14.3%; *p* < 0.001), while the incidence of perineural invasion was similar between the two groups (see Table 1). Besides, the clinicopathological characteristics of patients in the ESCC and E-NECs groups before and after matching are shown in Table 2.

### 3.3. Composition of Neuroendocrine Neoplasms

E-NENs consisted of 50.5% pure NENs (*n* = 54, including 5.6% [3/54] NETs and 94.4% [51/54] NECs), 34.6% NmSCC (*n* = 37), 13.1% (*n* = 14) NENs mixed with adenocarcinoma (NmAC), and 1.8% (*n* = 2) NENs mixed with other carcinomas; the latter three types are collectively termed MiNENs (Figure 2). NECs consisted of 70.6% (36/51) small-cell NECs (SCNECs), 27.4% (14/51) large-cell NECs (LCNECs), and 2.0% (1/51) mixed NECs. NETs were characterized by well-differentiated and medium-sized cells with abundant cytoplasm growing in the stroma in a nest form (Figure 3A). SCNECs consisted of small and round, ovoid, or spindle-like cells containing hyperchromatic nuclei with dense chromatin; meanwhile, LCNECs were composed of large cells with obvious nucleoli and eosinophilic cytoplasm (Figure 3B,C). MiNENs were primarily composed of NECs mixed with SCC or AC (Figure 3D,E); other carcinomas were rare.

### 3.4. Survival Analysis

The Kaplan–Meier analysis of survival rates showed that E-NECs were associated with poorer OS and RFS compared with ESCC (five-year OS: 35.4% vs. 54.8%, *p* = 0.0019; five-year RFS: 29.3% vs. 48.9%, *p* < 0.001). In the subset analysis, the survival disadvantage was prominent in tumors of all stages with the exception of stage I (Figure 4A–F). According to the different components of E-NECs, no significant difference in OS or RFS was observed between NmSCC and ESCC (Figure 5A,C). Although there was no statistical difference in OS between pNECs and NmSCC, the survival curves showed some separation. The RFS was significantly lower for pNECs than for NmSCC (Figure 5A,C). Moreover, pNECs showed significantly lower OS (30.7% vs. 54.8%, *p* < 0.001) and RFS (16.6% vs. 48.9%, *p* < 0.001) compared with ESCC. Apparently, the dismal survival associated with pNECs is more pronounced than that for overall E-NECs (Figure 5B,D). E-NECs were divided into two subgroups (surgery alone and multimodal therapy) based on the treatment modes, and the OS and RFS were compared between the subgroups. Surgery combined with neoadjuvant or adjuvant therapy did not significantly improve OS or RFS compared with surgery alone; only patients with stage III–IV tumors showed a better OS (Figure 6A–D).

In the univariate analysis, E-NECs were associated with poor OS (after matching: hazard ratio [HR] = 1.877, 95% confidence interval [CI] = 1.253–2.810, *p* = 0.002) and RFS (after matching: HR = 1.929, 95% CI = 1.321–2.817, *p* = 0.003). In the multivariate analysis, E-NECs were an independent predictive factor for poor OS (after matching: HR = 2.008, 95% CI = 1.338–3.013, *p* = 0.001) and RFS (after matching: HR = 2.109, 95% CI = 1.435–3.099, *p* < 0.001). All results are shown in Table 3. Besides, the hazard ratios of other factors associated with OS and RFS are shown in Appendix A.

### 3.5. Lymphocyte Infiltration

The number of CD8^+^ T cells in the TN differed significantly between E-NECs and ESCC (128/mm^2^ vs. 568/mm^2^, *p* = 0.032), and the difference was more obvious between pNECs and ESCC (48/mm^2^ vs. 568/mm^2^, *p* = 0.002). The number of CD4^+^ T cells in the TN was significantly lower for E-NECs than for ESCC (185/mm^2^ vs. 567/mm^2^, *p* = 0.007), and the difference was more prominent between pNECs and ESCC (110/mm^2^ vs. 567/mm^2^, *p* = 0.021; Figure 7A). In the TS, the number of CD8^+^ T cells was not significantly different between E-NECs and ESCC (887/mm^2^ vs. 1143/mm^2^, *p* = 0.125); however, the number of CD4^+^ T cells was lower for E-NECs than for ESCC (1545/mm^2^ vs. 2872/mm^2^, *p* = 0.013). The numbers of CD8^+^ and CD4^+^ T cells were not statistically different between pNECs and ESCC (CD8^+^ T cells: 1202/mm^2^ vs. 1143/mm^2^, *p* = 0.872; CD4^+^ T cells: 1549/mm^2^ vs. 2872/mm^2^, *p* = 0.06, see Figure 7B). While the number of CD20^+^ B cells was not significantly different between E-NECs and ESCC (1635/mm^2^ vs. 3470/mm^2^, *p* = 0.053), pNECs were associated with fewer CD20^+^ B cells than ESCC (1163/mm^2^ vs. 3470/mm^2^, *p* = 0.021; Figure 7C). Figure 8 shows the fields of view (40× magnification) used to quantify the TILs in the E-NECs and ESCC groups.

## 4. Discussion

NENs originate from the neuroendocrine system. They can be present throughout the body, although the vast majority are located in the digestive system and lungs. Gastroenteropancreatic neuroendocrine neoplasms (GEP-NENs) are the most common type of NENs in the digestive tract, accounting for approximately 55–70% of NENs [8]. Improvements in diagnosis and physical examination have led to increasing incidence. A population-based study found that the annual incidence of NENs is on the rise [5]. E-NENs are highly heterogeneous tumors, among which E-NECs are characterized by high malignancy and aggressiveness, along with a poorer prognosis compared with other common tumors [14]. According to the pathological outcomes, advanced tumors are more common in E-NENs compared with ESCC, which may explain the poorer prognosis of E-NENs. NENs can produce peptide hormones that lead to carcinoid syndromes such as flushing. NENs also appear in the EC, although their incidence is lower than ESCC and AC, and E-NENs rarely secrete hormones [8]. E-NECs have accounted for approximately 2.5–5.9% of all EC over the past 20 years [15], and the incidence of E-NENs ranges from approximately 0.8–2.8% [16,17]. In our study, E-NENs accounted for 3%, in accordance with past findings.

According to the latest World Health Organization classification of tumors of the digestive system, NENs can be divided into NETs, NECs, and MiNENs. NETs are well-differentiated tumors that can be graded as G1, G2, or G3 based on tumor proliferation. NECs include SCNECs and LCNECs, while MiNENs include NmSCC, NmAC, and NENs mixed with other carcinomas. Mastracci et al. reported that E-NENs mainly consist of NECs and MiNENs, while SCNECs account for the majority of NECs [18]. The vast majority of MiNENs are NmSCCs in Asia, whereas NmACs are predominant in Western countries [19]. This difference may be related to genes and the environment. In the present study, NECs and MiNENs accounted for 47.7 and 49.5% of the cases, respectively, while NETs accounted for only 2.8%. Among NEC cases, the most common types were SCNECs (36/51) and LCNECs (14/51); there was only one case of mixed NEC containing both small- and large-cell carcinoma. MiNENs included 69.8% (37/53) NmSCCs, 26.4% (14/53) NmACs, and 3.8% (2/53) NENs mixed with other carcinomas, which is in line with the characteristics of the Asian population. Similarly, Mura et al. [20] found better survival for MiNENs compared with pNENs and indirectly suggested a poor prognosis of NENs.

Compared with other common NENs, E-NENs have received little attention. Therefore, no treatment guidelines or consensuses are available for esophageal NENs. According to the NET guidelines, surgery is the primary treatment for NETs. However, the guidelines do not specifically introduce the treatment of esophageal NENs but rather describe the common NENs [21,22]. Recently, with the increase in cases, therapeutic experiences with E-NENs have gradually accumulated. At present, the primary treatment for E-NENs is multidisciplinary therapy, including surgery, CT, RT, biological therapy, and target therapy. Surgery is usually the only radical measure. Similar to ESCC, radical esophagectomy, which involves the removal of the primary tumor and dissection of regional lymph nodes, is preferred for resectable tumors. However, surgery cannot provide a survival benefit for all E-NENs; its effect depends on the tumor stage. Limited-stage tumors are confined to primary sites without regional lymph node or distant metastasis, unlike advanced-stage disease. The therapeutic efficacy of radical esophagectomy is better than CT or/and RT and is the first option for limited-stage E-NENs [9,15]. A retrospective study based on a large database showed that surgery combined with CT or RT resulted in better OS and cancer-specific survival (CSS) compared with CT alone, RT alone, or chemoradiotherapy (CRT) [23]. Even so, the postoperative recurrence risk of E-NENs was still higher compared with ESCC, and the median OS was significantly lower [24], especially in patients with lymph-node metastasis. In contrast, a Japanese multi-center retrospective study found no significant difference in OS between CRT and surgery, and surgery was even associated with poorer OS for stage III–IV disease [10]. Meng et al. reported that CRT was associated with better survival than surgery plus CT or RT for limited-stage SCNEC [25].

We observed a prominent difference in OS and RFS of between E-NECs and ESCC, with the exception of the stage I subgroup. The findings indicate that perhaps only early-stage patients are good candidates for surgery. Xu et al. reported that surgery should be recommended as a primary treatment modality for stage I NECs [14]. Hence, preoperative diagnosis is particularly important. Multiple biopsies or bite-on-bite biopsy should be considered for precise diagnosis. If histological typing indicates E-NENs, the clinical stage should be carefully assessed. However, these suggestions need to be verified through large-scale randomized controlled trials, and surgery-based treatments require further exploration for patients with early-stage E-NENs.

For patients with advanced-stage tumors, surgery usually aims to reduce symptoms and tumor burden. CRT may be the best choice for unresectable disease; it can prolong survival as much as possible [8]. In addition to traditional therapy, peptide receptor radionuclide therapy can be implemented for some E-NETs that can release hormones and express somatostatin receptors (SSRs). The radionuclide-labeled somatostatin analogues (SSAs) can relieve symptoms by binding the receptors and inhibiting tumor growth via internal irradiation. SSAs exert anti-proliferative activity by stimulating SSRs to mediate apoptosis [26,27]. Radiofrequency ablation and transcatheter chemical embolism are optional treatment measures for unresectable E-NETs (e.g., patients with liver metastasis). We found that multimodal therapy did not significantly improve the survival duration for E-NECs, suggesting that current traditional treatments are inadequate. Thus, more effective, novel, and combined therapies are urgently needed.

As a key part of the immune system, lymphocytes play an important role in anti-tumor immunity. Increasing evidence suggests that TILs are important predictive markers for prognosis. TILs mainly consist of T cells, B cells, and little natural killer cells [28,29]. Among them, CD8^+^ T cells are also called cytotoxic T lymphocytes (CTLs) and can kill tumor cells directly. In a meta-analysis, Gao et al. demonstrated that a high level of CD8^+^ T cells was associated with better OS [30]. CD4^+^ T cells can either promote the function of CTLs or inhibit the function of immune cells [31]. Despite this bidirectional effect, patients with high levels of CD4^+^ T cells showed significantly improved OS and RFS [30,32]. For ESCC, the high infiltration of CTLs in the TN was found to be associated with favorable OS [33], and the outcome was indirectly observed in our study. This indicates that CTLs play a crucial role in anti-tumor immunity for ESCC. In the present study, the numbers of both CD8^+^ and CD4^+^ T cells in the TN were significantly lower for E-NENs than for ESCC, and the difference was especially prominent for CD8^+^ T cells. Thus, the lack of sufficient CTLs may explain the dismal prognosis for E-NENs. In the subset analysis, pNECs were associated with poorer survival duration and fewer TILs than E-NECs overall, further confirming the correlation between prognosis and TILs.

The programmed death protein-1 (PD-1)/programmed death-ligand 1 (PD-L1) immune checkpoint pathway has attracted considerable attention. The goal of immunotherapy is to generate an effective CTL response, which can be realized by reactivating pre-existing tumor-specific CTLs and/or stimulating naive CD8^+^ T cells [34,35,36]. In other words, the survival benefit conferred by immune checkpoint inhibitors (ICIs) can be attributed to the CTL response [37]. ICIs have achieved favorable efficacy for Merkel cell carcinoma, a type of NEN originating from the skin; and this tumor, with a high level infiltration of CD8^+^ T cells, significantly responds to ICIs, and more improved clinical outcomes have been observed [38,39]. Moreover, the phenomena that a high level of immune cell infiltration is correlated with the response to ICIs has been observed in other cancers [40,41]. Therefore, we assumed that less infiltration of CD8^+^ T cells may lead to the suboptimal efficacy of immunotherapy for E-NENs.

At present, ICIs have obtained good efficacy in many solid tumors. However, immunotherapy for NENs remains in the exploratory phase. Although most clinical trials on immunotherapy for NENs have been phase II, single-arm studies, the overall preliminary results are promising [42]. Unfortunately, no clinical trials of E-NENs have been reported. Considering the favorable efficacy of ICIs in many malignant tumors, immunotherapy remains an important aspect in exploring novel therapeutic methods for E-NENs. The PD-L1 expression of poorly differentiated NECs was higher than that for well-differentiated NETs, and NECs tend to have a higher objective response rate compared with NETs [42]. Additionally, the tumor mutation burden (TMB) is typically lower in NETs and higher in NECs [42]. In many cancers, higher PD-L1 expression and TMB predict a better prognosis and response to ICIs [43]. Given that most E-NENs are poorly differentiated carcinomas, perhaps the beneficiaries of immunotherapy could be selected.

When immunotherapy is considered for patients with E-NENs, immune activation strategies that make E-NENs more susceptible to ICIs need to be explored and developed in future research. Sequential CT is beneficial for increasing the TMB. Prior to immunotherapy, radiotherapy induces inflammation in the tumor microenvironment that augments TILs [24]. The underlying molecular mechanisms of the cold tumor phenotype of E-NENs are deserving of more consideration. On the other hand, selecting the patients that can benefit from immunotherapy and identifying molecular markers that can predict the response to immunotherapy (e.g., the TMB) are also important. New therapies combined with useful markers are thus the most important focus of clinical studies for E-NENs.

Our study has some limitations. First, this was a single-center, retrospective study, and the results need to be verified in a prospective, multi-center study. Second, the stage-I subgroup had few cases, and the hypothesis that surgery is more suitable for early-stage patients was not validated by large-scale, prospective data. Third, the analysis results were mainly from E-NECs patients due to the rarity and special biological behavior of NET cases. Moreover, we did not directly compare survival between surgery and other treatment methods. Therefore, the optimal treatment mode for E-NENs requires additional exploration. Finally, the relationship between TILs and prognosis was not directly confirmed, which needs to verified by an evaluation of immune infiltration with a large-scale sample.

## 5. Conclusions

With the exception of early-stage disease, E-NECs are significantly correlated with a poorer prognosis than ESCC. The weaker anti-tumor immunity caused by the presence of fewer TILs within the tumor immune microenvironment may contribute to the poor prognosis of E-NECs. For advanced-stage E-NECs, conventional therapies are inadequate, and more intensive and novel therapeutic measures are warranted.

## Figures and Tables

**Figure 1 cancers-15-01732-f001:**
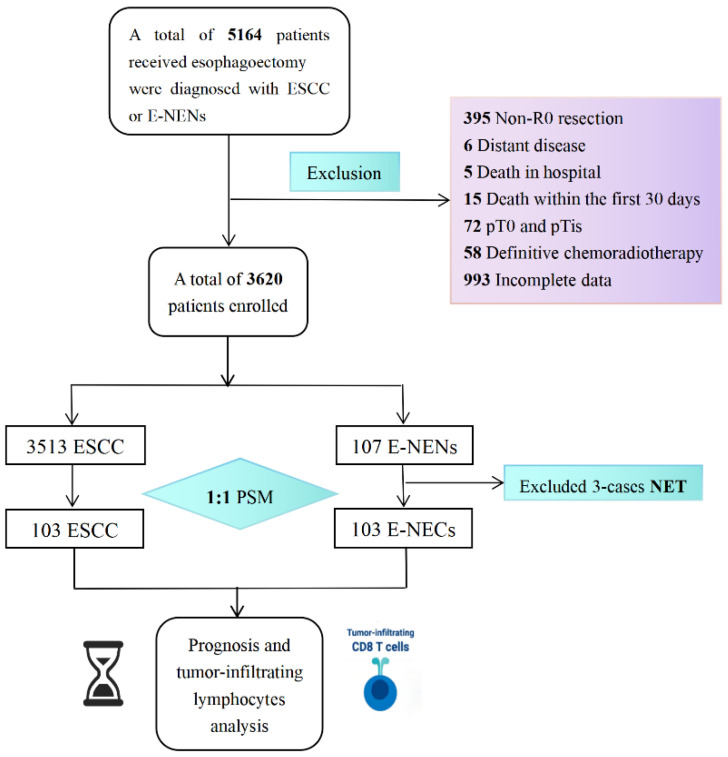
The study flowchart of enrolled population.

**Figure 2 cancers-15-01732-f002:**
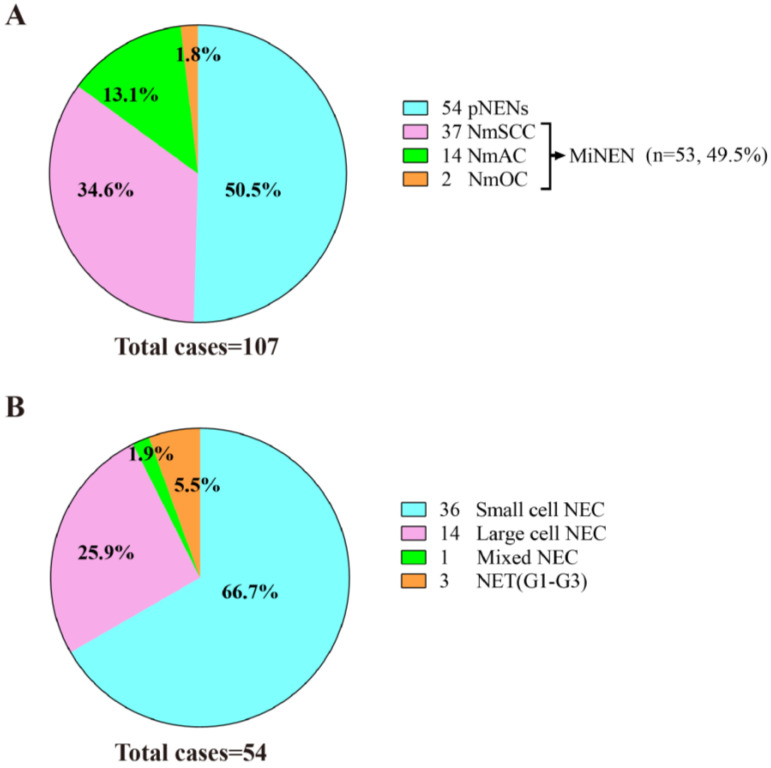
Pie chart of the number of E-NENs patients. (**A**) The constituent proportion of all patients with E-NENs. (**B**) The composition ratio of pNENs patients. Abbreviations: pNENs, pure neuroendocrine neoplasms; NmSCC, neuroendocrine neoplasms mixed with squamous cell carcinoma; NmAC, neuroendocrine neoplasms mixed with adenocarcinoma; NmOC, neuroendocrine neoplasms mixed with other carcinomas; NEC, neuroendocrine carcinoma; NET, neuroendocrine tumor.

**Figure 3 cancers-15-01732-f003:**
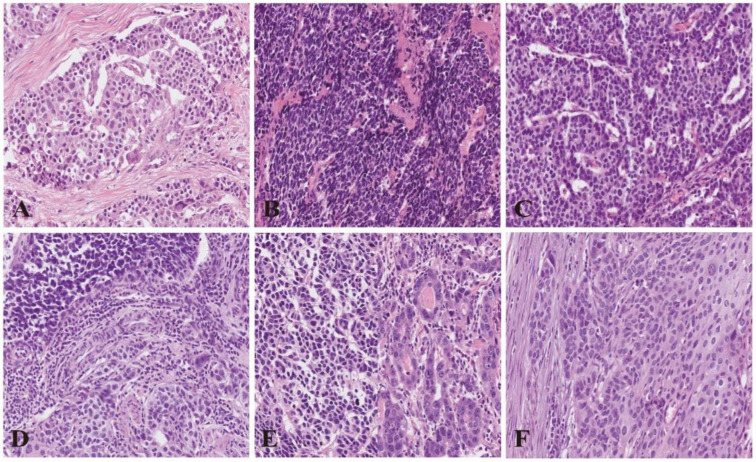
Cell morphology of esophageal neuroendocrine neoplasms and esophageal squamous cell carcinoma under 40× magnification using hematoxylin and eosin staining: (**A**) neuroendocrine tumor; (**B**) small-cell neuroendocrine carcinomas (NECs); (**C**) large-cell NECs; (**D**) small-cell NECs mixed with squamous cell carcinoma; and (**E**) small-cell NECs mixed with adenocarcinoma; and (**F**) esophageal squamous cell carcinoma.

**Figure 4 cancers-15-01732-f004:**
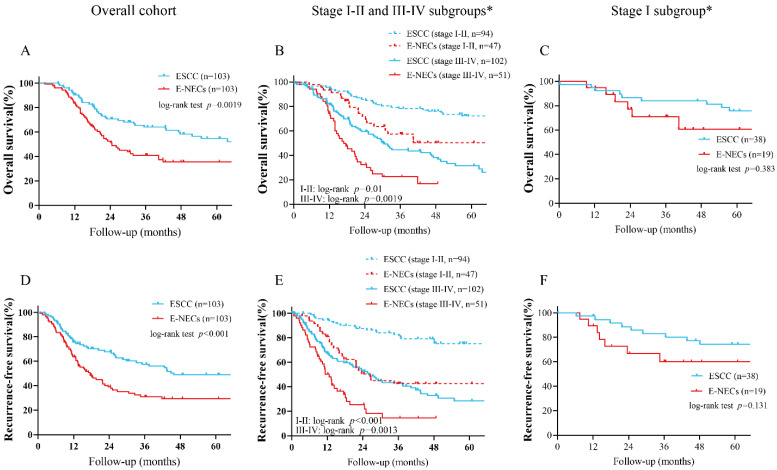
Kaplan–Meier analysis of OS and RFS for the overall cohort and subsets: (**A**,**D**) OS and RFS curves for the overall cohort; (**B**,**E**) OS and RFS curves of patients in the stage I–II and III–IV subgroups; (**C**,**F**) OS and RFS curves of patients in the stage I subgroup. Abbreviations: OS, overall survival; RFS, recurrence-free survival; E-NENs, esophageal neuroendocrine neoplasms; ESCC, esophageal squamous cell carcinoma. * The stage I, I–II, and III–IV subgroups were formed by the PSM with 1:2 ratio; namely, these 3 cohorts were generated from 3-times independent PSM.

**Figure 5 cancers-15-01732-f005:**
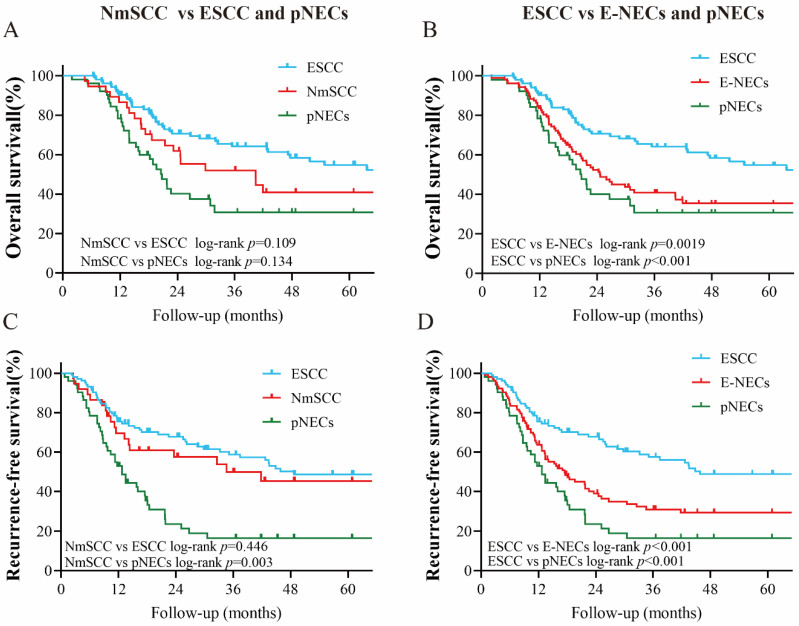
Comparisons of OS and RFS between subgroups of E-NECs with different components: (**A**,**C**) NmSCC vs. ESCC and pNECs; (**B**,**D**) ESCC vs. E-NECs and pNECs. Abbreviations: ESCC, esophageal squamous cell carcinoma; E-NECs, esophageal neuroendocrine carcinomas; pNECs, pure neuroendocrine carcinomas; NmSCC, neuroendocrine neoplasms mixed with squamous cell carcinoma.

**Figure 6 cancers-15-01732-f006:**
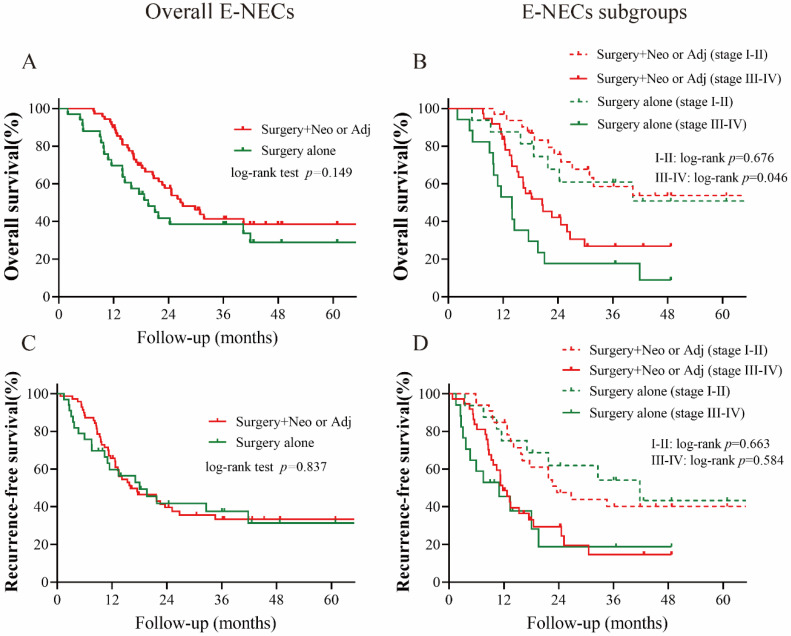
Comparisons of OS and RFS between subsets of E-NECs: (**A**,**C**) surgery alone vs. surgery combined with Neo or Adj in all E-NECs cases; (**B**,**D**) surgery alone vs. surgery combined with Neo or Adj in the stage I–II and III–IV subgroups. Abbreviations: E-NECs, esophageal neuroendocrine carcinomas; OS, overall survival; RFS, recurrence-free survival; Neo, neoadjuvant therapy; Adj, adjuvant therapy.

**Figure 7 cancers-15-01732-f007:**
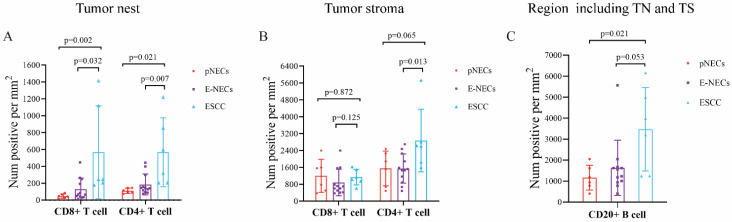
Comparisons of the number of TILs in the TN and TS between E-NECs (including the pNECs subgroup) and ESCC: (**A**,**B**) comparison of the numbers of CD8^+^ T and CD4^+^ T cells in the TN and TS between ESCC and E-NECs; (**C**) comparison of the number of CD20^+^ B cells between ESCC and overall E-NECs. Abbreviations: E-NECs, esophageal neuroendocrine carcinomas; pNECs, pure neuroendocrine carcinomas; ESCC, esophageal squamous cell carcinoma; TILs, tumor-infiltrating lymphocytes; TN, tumor nest; TS, tumor stroma.

**Figure 8 cancers-15-01732-f008:**
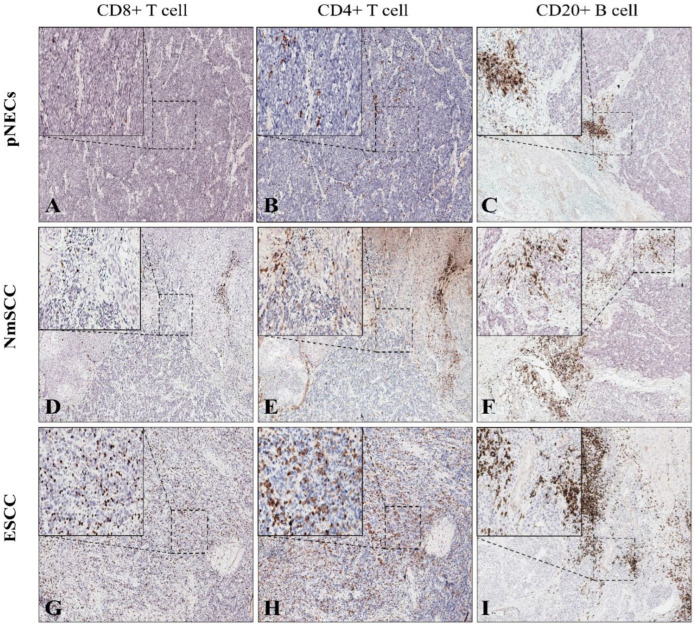
Identification of TILs using immunohistochemical staining under a 400× high-power field: detection of CD4^+^ T, CD8^+^ T, and CD20^+^ B cells in the (**A**–**C**) pNECs subgroup; (**D**–**F**) NmSCC subgroup; and (**G**–**I**) ESCC group. Abbreviations: pNECs, pure neuroendocrine carcinomas; NmSCC, neuroendocrine neoplasms mixed with squamous cell carcinoma; ESCC, esophageal squamous cell carcinoma; TILs, tumor-infiltrating lymphocytes.

**Table 1 cancers-15-01732-t001:** Baseline characteristics of the enrolled population (including 3 NET cases).

Baseline Characteristics	ESCC	E-NENs	*p*-Value
	3513 (n (%))	107 (n (%))	
Sex			0.116
Male	2928 (83.3%)	83 (77.6%)	
Female	585 (16.7%)	24 (22.4%)	
Age (years, median = 64)			0.001
≥64	1813 (51.6%)	72 (67.3%)	
<64	1700 (48.4%)	35 (32.7%)	
Smoker			0.873
No	1483 (42.2%)	46 (43.0%)	
Yes	2030 (57.8%)	61 (57.0%)	
Drinking			0.658
No	1652 (47.0%)	48 (44.9%)	
Yes	1861 (53%)	59 (55.1%)	
BMI (kg/m^2^, median (IQR))	22.8 (20.7–24.8)	23.2 (21.2–25.9)	0.126
Lesion size (cm, median = 3.5)			0.487
<3.5	1686 (48.0%)	55 (51.4%)	
≥3.5	1827 (52.0%)	52 (48.6%)	
Tumor location			0.018
Upper	456 (13.0%)	7 (6.5%)	
Middle	1860 (52.9%)	51 (47.7%)	
Lower	1197 (34.1%)	49 (45.8%)	
T status *			<0.001
T0 + T1 + T2	1579 (44.9%)	68 (63.6%)	
T3 + T4	1934 (55.1%)	39 (36.4%)	
N status *			<0.001
N0	1864 (53.1%)	39 (36.5%)	
N1	962 (27.4%)	29 (27.1%)	
N2	519 (14.8%)	27 (25.2%)	
N3	168 (4.7%)	12 (11.2%)	
Tumor stage *			0.011
I	749 (21.3%)	22 (20.6%)	
II	1221 (34.8%)	29 (27.1%)	
III	1340 (38.1%)	42 (39.2%)	
Iva	203 (5.8%)	14 (13.1%)	
Tumor differentiation			0.003
Well	432 (12.3%)	3 (2.8%)	
Moderately/poorly	3081 (87.7%)	104 (97.2%)	
LVI			<0.001
Negative	3012 (85.7%)	78 (72.9%)	
Positive	501 (14.3%)	29 (27.1%)	
PNI			0.956
Negative	3146 (89.6%)	96 (89.7%)	
Positive	367 (10.4%)	11 (10.3%)	
Neoadjuvant therapy			0.006
No	3023 (86.1%)	102 (95.3%)	
Yes	490 (13.9%)	5 (4.7%)	
Adjuvant therapy			<0.001
No	1814 (51.6%)	35 (32.7%)	
Yes	1699 (48.4%)	72 (67.3%)	

Abbreviations: ESCC, esophageal squamous cell carcinoma; E-NENs, esophageal neuroendocrine neoplasms; BMI, body mass index; IQR, interquartile range; LVI, lymphovascular invasion; PNI, perineural invasion. * T status, N status, and tumor stage included pathological category (pT, pN, pStage) and category (ypT, ypN, ypStage) after neoadjuvant therapy.

**Table 2 cancers-15-01732-t002:** Clinicopathological characteristics of patients in the two groups (excluding 3 cases of NETs) before and after propensity-score matching ^#^.

Characteristics	Before Matching			After Matching		
	ESCC	E-NECs	*p*-Value	ESCC	E-NECs	*p*-Value
	3513 (No. [%)])	104 (No. [%])		103 (No. [%])	103 (No. [%])	
Sex			0.142			0.866
Male	2928 (83.3%)	81 (77.9%)		81 (78.6%)	80 (77.9%)	
Female	585 (16.7%)	23 (22.1%)		22 (21.4%)	23 (22.1%)	
Age (years, median (IQR))	63 (58–68)	66 (61–71)	0.003	65 (61–70)	66 (61–71)	0.988
Smoking			0.830			0.888
No	1483 (42.2%)	45 (43.3%)		46 (44.7%)	45 (43.7%)	
Yes	2030 (57.8%)	59 (56.7%)		57 (55.3%)	58 (56.3%)	
Drinking			0.712			0.210
No	1652 (47.0%)	47 (45.2%)		56 (54.4%)	47 (45.6%)	
Yes	1861 (53%)	57 (54.8%)		47 (45.6%)	56 (54.4%)	
BMI (kg/m^2^, median (IQR))	22.8 (20.7–24.8)	22.8 (21.2–25.2)	0.185	23.6 (21.4–25.7)	23.2 (21.2–25.3)	0.436
Lesion size (cm, median (IQR))	3.5 (2.5–4.5)	3.3 (2.0–5.0)	0.604	3.5 (2.7–4.5)	3.3 (2.0–5.0)	0.423
Location			0.028			0.959
Upper	456 (13.0%)	7 (6.7%)		7 (6.8%)	7 (6.8%)	
Middle	1860 (52.9%)	50 (48.1%)		52 (50.5%)	50 (48.5%)	
Lower	1197 (34.1%)	47 (45.2%)		44 (42.7%)	46(44.7%)	
T status *			<0.001			0.774
T0 + T1 + T2	1579 (44.9%)	66 (63.5%)		63 (61.2%)	65 (63.1%)	
T3 + T4	1934 (55.1%)	38 (36.5%)		40 (38.8%)	38 (36.9%)	
N status *			<0.001			0.815
N0	1864 (53.1%)	38 (36.6%)		38 (36.9%)	38 (36.9%)	
N1	962 (27.4%)	28 (26.9%)		23 (22.3%)	28 (27.2%)	
N2	519 (14.8%)	26 (25.0%)		30 (29.1%)	25 (24.3%)	
N3	168 (4.7%)	12 (11.5%)		12 (11.7%)	12 (11.6%)	
Tumor stage *			0.008			0.365
I	749 (21.3%)	21 (20.2%)		12 (11.7%)	21 (20.4%)	
II	1221 (34.8%)	28 (26.9%)		34 (33.0%)	28 (27.2%)	
III	1340 (38.1%)	41 (39.4%)		41 (39.8%)	40 (38.8%)	
IVa	203 (5.8%)	14 (13.5%)		16 (15.5%)	14 (13.6%)	
Tumor Differentiation			<0.001			1
Well	432 (12.3%)	0 (0%)		0 (0%)	0 (0%)	
Moderately/poorly	3081 (87.7%)	104 (100%)		103 (100%)	103 (100%)	
LVI			0.001			0.745
Negative	3012 (85.7%)	77 (74.0%)		79 (76.7%)	77 (74.0%)	
Positive	501 (14.3%)	27 (26.0%)		24 (23.3%)	26 (26.0%)	
PNI			0.784			0.298
Negative	3146 (89.6%)	94 (90.4%)		97 (94.2%)	93 (90.4%)	
Positive	367 (10.4%)	10 (9.6%)		6 (5.8%)	10 (9.6%)	
Neoadjuvant therapy			0.008			1
No	3023 (86.1%)	99 (95.2%)		98 (5.2%)	98 (95.2%)	
Yes	490 (13.9%)	5 (4.8%)		5 (4.8%)	5 (4.8%)	
Adjuvant therapy			<0.001			0.653
No	1814 (51.6%)	34 (32.7%)		31 (30.0%)	34 (32.7%)	
Yes	1699 (48.4%)	70 (67.3%)		72 (70.0%)	69 (67.3%)	

Abbreviations: ESCC, esophageal squamous cell carcinoma; E-NECs, esophageal neuroendocrine carcinomas; NET, neuroendocrine tumor; BMI, body mass index; IQR, interquartile range; LVI, lymphovascular invasion; PNI, perineural invasion. * T status, N status, and tumor stage included pathological category (pT, pN, pStage) and category (ypT, ypN, ypStage) after neoadjuvant therapy. ^#^ Significant differences were identified in age, tumor location, T status, N status, differentiation, LVI, and neoadjuvant and adjuvant therapy. These variables were used for matching.

**Table 3 cancers-15-01732-t003:** Univariate and multivariate analysis of the histology related to overall survival and recurrence-free survival ^#^.

Survival Status	Before Matching						After Matching					
	Univariate Analysis			Multivariate Analysis			Univariate Analysis			Multivariate Analysis		
OS	HR	95%CI	*p*-Value	HR	95%CI	*p*-Value	HR	95%CI	*p*-Value	HR	95%CI	*p*-Value
E-NECs vs. ESCC	1.995	1.536–2.591	<0.001	1.897	1.453–2.477	<0.001	1.877	1.253–2.810	0.002	2.008	1.338–3.013	0.001
RFS												
E-NECs vs. ESCC	2.257	1.768–2.881	<0.001	2.105	1.640–2.702	<0.001	1.929	1.321–2.817	0.001	2.109	1.435–3.099	<0.001

Abbreviations: HR, hazard ratio; CI, confidence interval; ESCC, esophageal squamous cell carcinoma; E-NECs, esophageal neuroendocrine carcinomas; OS, overall survival; RFS, recurrence-free survival. ^#^ The hazard ratios for OS and RFS were calculated based on the cohort excluding the 3 cases of NETs.

## Data Availability

The study data were obtained from the Shanghai Esophageal Cancer Cohort Database of Shanghai Hospital Development Center.

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
