# Peer review of "Analysis of the Clinicopathological Characteristics, Prognosis, and Lymphocyte Infiltration of Esophageal Neuroendocrine Neoplasms: A Surgery-Based Cohort and Propensity-Score Matching Study"

_cancers, 2023, doi:10.3390/cancers15061732_

Round 1

Reviewer 1 Report

The vast majority of NENs were NECs. It would be better to exclude the well differentiated NETs, since they have different tumor biology compared to NECs.

There is no information about chemotherapy regimens and the adjuvant treatments are not described.

Table 3 is difficult to read. “Ref” is not explained.

Introduction: The treatment methods for E-NENs refer to the existing protocol for SCC or AC, including surgery, CT, radiotherapy (RT), interventional therapy, and biological therapy.” 

Is this correct? In that case, references are needed. 

Methods, 2.1 Study cohort: “Patients who did not obtain R0 resection…… were excluded. Patients who received radical chemoradiotherapy were not included in this study. “  

Why were these patients excluded?

Methods, 2.3 Surgical procedures: “Neoadjuvant or adju- vant therapy, typically including CT, RT, and immunotherapy, are recommended for patients with locally advanced tumor stages.” 

Please add reference

Methods, 2.6 Assessment of tumor-infiltrating lymphocytes: “We randomly selected paraffin sections of six cases with pure neuroendocrine neo- plasms (pNENs), NENs mixed with squamous cell carcinoma (NmSCC), and ESCC. “  

This is fairly few patients in each group.

Results, 3.1 Overall cohort characteristics: “The proportion of patients who received neoadjuvant or adjuvant therapies…” 

There is no information about the kind of adjuvant treatment including chemotherapy regimens

Results, 3.2 Pathological outcomes: “Lymph-node metastasis (63.5% vs. 46.9%, p < 0.001) and advanced tumors (stages III+IV, 52.3% vs. 43.9%, p = 0.011) were more common in E- NENs than in ESCC. In addition, the vast majority of E-NENs were poorly differentiated tumors.” 

This may explain the poorer prognosis for patients with NEN and should be commented on in the discussion.

Discussion, Improvements in diagnosis and physical examination have led to increasing morbidity. 

Those improvements have led to increasing incidence, but not to increasing morbidity

Discussion: “NENs are characterized by high malignancy and aggressiveness along with poorer prognosis compared with other common tumors [13].” 

This may be true for NECs, but not for NETs.

Discussion: “Due to the paucity of cases, no treatment guidelines or consensus are available for NENs, with the exception of pancreatic NENs.” 

This is wrong.

Discussion: “At present, the primary treatment for E-NENs is multidisciplinary therapy including surgery, CT, RT, biological therapy, and target therapy.” 

Please add reference.

Discussion: “Thus, the lack of sufficient CTLs may explain the dismal prognosis for E-NENs.” 

This may also be explained by the more frequent occurrence of lymph node metastases and more advance tumor stage in the NEN group.

Discussion:”… the overall preliminary results are promising [36].”

On the contrary, the authors state “In fact, with regard to the well-differentiated forms of NENs (NETs), the results obtained nowadays have been disappointing. “

Discussion: “In summary, when immunotherapy is considered for patients with E-NENs, immune activation strategies that make E-NENs more susceptible to ICIs need to be explored and developed in future research.”

Please rewrite. Remove “in summary”.

Conclusion: “The weaker anti-tumor immunity caused by the presence of fewer TILs within the tumor immune microenvironment may explain the poor prognosis of E-NENs.”

Please change to ““The weaker anti-tumor immunity caused by the presence of fewer TILs within the tumor immune microenvironment may contribute to the poor prognosis of E-NENs.”

Author Response

Dear reviewer,

Thank you for the feedback and the meaningful comments. We have carefully reviewed all comments, and we have tried our best to revise the manuscript. Our point-by-point responses to the comments are in the attachment as Word file. Please see the attachment.

Reviewer 2 Report

Zhang et al. performed a single-center retrospective study to investigate the differences in clinicopathological features, prognosis, and TILs between E-NENs and ESCC. The authors concluded that E-NEN were correlated with poorer prognosis than ESCC, except for early-stage cases, which may be due to less TIL infiltration. Since E-NEN is a rare disease and there are few previous reports, the findings presented in the manuscript are highly appreciated. My comments are as follows.

1.         Although propensity score matching was performed between the E-NEN and ESCC groups, further comparative analyses were performed as subgroup analyses. In such cases, background factors that were well balanced between the original two groups may become unbalanced between the subgroups. Consultation with a biostatistician is recommended.

2.         How many cases with hormone-induced symptoms were seen in the E-NEN group? If you have any information, please consider adding it.

3.         The authors described that TIL was evaluated in 6 cases each of pNEN, NmSCC and ESCC, what was the clinicopathological characteristics of them? Were the tumor stage and other background factors aligned?

4.         The description of immunotherapy in Discussion is redundant and a bit of a leap from the findings of TIL infiltration obtained in this study. Please consider reducing the amount of description.

5.         In “2.3. Surgical procedures”, what regimen was used for neoadjuvant or adjuvant chemotherapy?

6.         In “2.5. Outcomes follow-up” section, a range should be added for the median follow-up period.

7.         Parts of many tables are not displayed properly and should be corrected.

Author Response

Dear reviewer,

Thank you for the feedback and the meaningful comments. We have carefully reviewed all comments, and we have tried our best to revise the manuscript. Our point-by-point responses to the comments are in attachment as a Word file. Please see the attachment.

Reviewer 3 Report

The heading and summaries should mention that cases were selected from surgically resected cases, that the study is retrospective, and should also mention the total number of NEC, MiNEN and NET cases included.

In summaries: The number of specimens assessed for immune cell infiltrates should be stated (6 NEN, 6 MiNEN). It should be stated that only 4 patients had NET and therefore the conclusions do not apply to these. In reporting comparisons between treatments, indicate that these are nonrandomised.

The main conclusions are not sufficiently supported by data. Even if prognosis is poorer with resection of >stage I, a significant fraction obtained long-term survival and similar data on definitive chemo-radiotherapy was not presented. The second main conclusion was based on assement of immunophenotypes in only 12 heterogeneous NEN-cases, that were not taken from an unselected populaton, and the immunophenotype was not assessed for NET.

State that the contreversy regarding surgery only applies to NEC cases, not to NET (see current NET-guidelines).

Pure NET-cases should be excluded from calculations as they are biologically very different and no statistics can be done.

In the description of the study cohort indicate how many cases that were excluded due to individual causes.

What staging system is used? It is shown in tables that patients with stage IV disease are included - are they all stage IVa and radically resected?

The patients are not representing an unbiased sample of the whole population. NEC may be much more frequent among patients with inoperable disease or may be treated preferably with definitive chemoradiotherapy.

Staging is not done according to modern standards (PET/CT), NET should be staged according to current guidelines.

The oncological treatment is not described sufficiently. The postoperative stage in patients receiving preoperative treatment is not prognostically comparable with that of patients, who were operated upfront. A major flaw is that survival was calculated from surgery, introducing a systematic bias (of +2 months?) for patients receiving preoperative treatment.

In assessing immune cell infiltrates, why not include all possible cases? Why not investigate PD-L1 expression?

Discussion:

References are not up to date. There are now a number of NET-guidelines.

Chemo-radiotherapy is not indicated for distant disease. PRRT is only relevant for NET. The use of RFA and chemoembolisation is not evidence based for local treatment of esophageal tumors, but may be used for NET liver metastases (outside the scope).

There are several reports in other tumors of associations between tumor immune cell infiltrates and effect of immunotherapy.

Author Response

Dear reviewer,

Thank you for the feedback and the valuable comments. We have carefully reviewed all comments from the reviewers, and we have tried our best to revise the manuscript. Our point-by-point responses to the comments are in the attachment. Please see the attachment.

Round 2

Reviewer 1 Report

Section 2.4, Diagnosis and classification: ”… with respective histological codes of 8240/3, 8246/3, and 8154/3.”. This can be removed.

Section 2.4, Diagnosis and classification: ”NET with well differentiated was graded into G1, G2, and G3 based on mitotic number and Ki-67 proliferation index”. Please add “according to the WHO classification year + reference.”

Section 2.4, Diagnosis and classification. Table1 should be removed. 

Section 2.6, Assessment of tumor-infiltrating lymphocytes (TILs): Why were no NENs mixed with adenocarcinoma examined? And the number of analyzed cases is small.

Section 3.4, Survival analysis: There are a lot of data, figures and tables in this section, which should be presented in a more easy accessible format.

Section 3.5. Lymphocyte infiltration: “In the TS, the number of CD8+ T cells was not

significantly different between E-NENs and ESCC” I suppose the authors mean In the TS, the number of CD8+ T cells was not significantly different between E-NECs and ESCC”

Section 3.5. Lymphocyte infiltration: “…however, the number of CD4+ T cells was lower for E-NECNs than for ESCC (1545/mm2 vs. 2872/mm2, = 0.013). The numbers of CD8+ and CD4+ T cells were not statistically different between pNENs and ESCC” I find this somewhat contradictory.

Section 4, Discusssion: “NENs are highly heterogeneous tumors, among

which NECs were characterized by high malignancy and aggressiveness along with

poorer prognosis compared with other common tumors [13].” This reference is about esophageal NECs. Please change the text.

Section 4, Discussion: “If histological typing indicates E-NECNs, the clinical stage should be carefully assessed “ Staging is important in almost all NENs and in all NECs, including all E-NENs and E-NECs.

Author Response

Dear reviewer,

Thank you for the valuable comments. We have carefully reviewed all comments and revised the manuscript.  Our point-by-point responses to the comments are shown in response letter.

Reviewer 2 Report

The authors have significantly improved the manuscript by addressing the reviewers’ comments. However, there are still several issues.

1.       Since the survival analysis as well as TIL infiltration assessment was finally performed only for NEC cases, NETs should be excluded before PS matching, given the difference in biological behavior. The authors should at least mention why they did not include the three NETs in the survival analysis.

2.       English editing should be done again by native speakers. For example, “Moreover, immunotherapy and chemotherapy as neoadjuvant treatment for partial patients.” contains a grammatical error.

Author Response

(The authors gave the same response as above.)

Reviewer 3 Report

The discussion of potential treatment options for NET should be deleted as there are only 3 of these cases included in the study.

Also the discussion regarding TIL and immunotherapy should be shortened to approx. 1/3, to reflect the limited information obtainable from the few cases assessed for TIL in the present study. 

The reference to Öberg from 2010 should be replased by a more recent review or guideline on neuroendocrine tumors.

Author Response

(The authors gave the same response as above.)

Round 3

Reviewer 1 Report

Introduction, Esophageal NENs (E-NENs) are a rare subtype of EC with poorly reported clinical and oncologic characteristics.”  What do you mean?

Results, Survival analysis, 3.4,: “E-NECsNs were divided into two subgroups (surgery alone and multimodal therapy) based on the treatment modes, and the OS and RFS were compared between the subgroups”. There is no detailed description about the adjuvant treatments given.